# Possibilities and limitations of antisense oligonucleotide therapies for the treatment of monogenic disorders

Marlen C. Lauffer [1], Willeke van Roon-Mom[1], Annemieke Aartsma-Rus[1✉] & N = 1 Collaborative*

Antisense oligonucleotides (ASOs) are incredibly versatile molecules that can be designed to specifically target and modify RNA transcripts to slow down or halt rare genetic disease progression. They offer the potential to target groups of patients or can be tailored for individual cases. Nonetheless, not all genetic variants and disorders are amenable to ASO-based treatments, and hence, it is important to consider several factors before embarking on the drug development journey. Here, we discuss which genetic disorders have the potential to benefit from a specific type of ASO approach, based on the pathophysiology of the disease and pathogenic variant type, as well as those disorders that might not be suitable for ASO therapies. We further explore additional aspects, such as the target tissues, intervention time points, and potential clinical benefits, which need to be considered before developing a compound. Overall, we provide an overview of the current potentials and limitations of ASO-based therapeutics for the treatment of monogenic disorders.

Antisense oligonucleotides (ASO) are short oligonucleotides that can bind to RNA in a target-specific manner and ultimately modify protein expressions. These molecules have shown incredible potential in the treatment of genetic disorders and can drastically alter the course of heritable diseases[1]. By interacting with RNA transcripts in a sequence-specific manner, ASOs can reduce target RNA transcript levels leading to limited expression of the encoding toxic proteins or can alter transcript sequences through splice-modulation to restore protein function where otherwise function would be lost[2]. These approaches have successfully been employed for multiple genetic disorders, such as spinal muscular atrophy (SMA), homozygous familial hypercholesterolemia, and primary hyperoxaluria type 1. Up to date, 15 oligonucleotides have so far received market authorization in different countries[3,4], with several others currently being tested in clinical phase I–III trials, with some edging closer to market approval[5,6].

ASOs are of special interest for approximately 8000 rare disorders, where rare by definition is a disease that affects less than 1 in 2000 people in Europe or less than 1 in 200,000 people in the US[7,8]. The high unmet medical need for developing treatments for rare disease patients is highlighted by the fact that no targeted therapeutics are available for over 90% of these disorders[9]. Since around 70–80% of rare disorders[9] are caused by pathogenic genetic alterations in single genes, ASO-based drugs could provide potentially disease-modifying therapies for many of these conditions. The recent successes of ASO-based therapies, especially the development of individualized treatments and treatments for disorders with only a few known cases, have given hope to the rare disease community[10,11].

[1] Dutch Center for RNA Therapeutics, Department of Human Genetics, Leiden University Medical Center, Leiden, The Netherlands. *A list of authors and their affiliations appears at the end of the paper. ✉email: a.m.rus@lumc.nl

Depending on the disease-causing mechanism, different ASO approaches may be applicable. As not all genetic diseases are amenable to ASO-based therapy, it is important to consider a multitude of aspects when deciding if an ASO is the most suitable therapeutic option for a given disease. This can be a complex process, and it is easy to overestimate clinical potential. In this regard, it is important to realize the limitations of the different ASO-based strategies, ensure transparent communication with affected individuals and their caregivers, and manage the hopes and expectations that come with new and often experimental treatments.

Here, we provide important considerations in determining the suitability of ASO approaches for different monogenic diseases. Besides the genetic background and pathophysiological mechanism of disease, we take the target tissue, delivery route, timing of intervention, and clinical outcome measures into account. We discuss the types of monogenic disorders that are treatable with the available ASO strategies and point out any potential hurdles that will need overcoming to expand the group of diseases that could become amenable to an ASO intervention.

## Mechanisms of action

ASOs are small, single- or double-stranded oligonucleotides that have been chemically modified to increase stability, improve the target affinity and bioavailability, and enhance cellular uptake. Depending on their chemical modifications, ASOs are subdivided into three generations[12]. ASOs can bind to their target RNA transcript in a sequence-specific manner via Watson–Crick base pairing. They can thus be designed to target distinct sequences or specific genetic variants[2]. Through this complementary binding, the molecules can alter protein expression by either knocking down transcripts or modifying pre-mRNA splicing, ultimately leading to either a reduction, modification, or restoration of a specific protein. According to their mechanism of action, ASOs can be divided into gapmer antisense oligonucleotides (gapmer ASOs) and small interfering RNAs (siRNAs) with which transcript knockdown can be achieved, and splice switching ASOs (ssASOs) used to modulate pre-mRNA splicing[2] (Fig. 1). ASOs that interfere with microRNAs, such as agomirs and antagomirs, are not considered here as these do not address the root cause of monogenic diseases.

**Transcript knockdown.** RNA transcripts can be knocked down using gapmer ASOs or siRNAs. Gapmers are single-stranded ASOs consisting of a DNA core flanked by modified RNA-based structures resistant to RNase H. Gapmers can be designed to target pre-mRNAs and mRNAs. Sequence-specific binding of the gapmer creates a DNA:RNA hybrid of the gapmer core with the target RNA transcript, which leads to the recruitment of RNase H, an enzyme that can recognize the DNA:RNA hybrids. RNase H cleaves the duplex structure, causing mRNA degradation and reducing gene expression[13] (Fig. 1A).

siRNAs, on the other hand, make use of the endogenous RNA interference pathway. These double-stranded RNA molecules occur naturally as microRNAs but can also be synthesized for a specific target (siRNA). Within the cell, the double-stranded siRNA is incorporated into the RNA-induced silencing complex (RISC), consisting of multiple proteins. The siRNA gets unwound and separated into the sense and antisense strands, with only the antisense strand remaining within the complex. The antisense strand then binds to its complementary mRNA target, mediating cleavage and leading to degradation of the RNA molecule[6] (Fig. 1B).

Compared to gapmer ASOs, siRNAs generally provide more efficient downregulation but, at the same time, have a higher

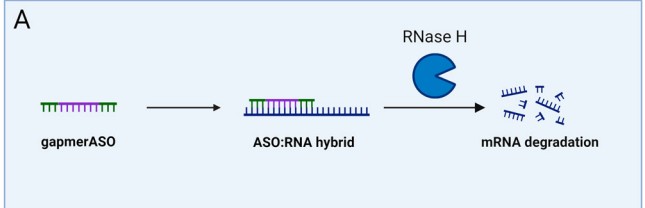

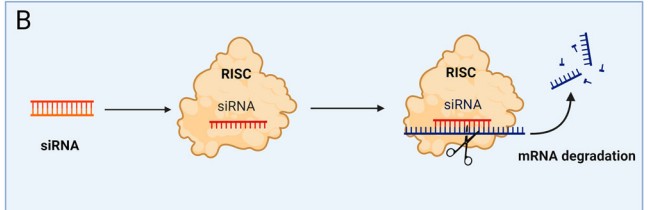

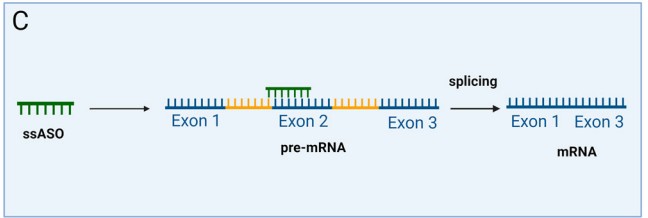

**Fig. 1 Overview of the action mechanisms of antisense oligonucleotides (ASOs). A** gapmer ASOs bind to their target mRNA to form DNA:RNA hybrids that recruit RNase H for the cleavage of mRNA. **B** siRNAs enter the cell as double strands and are incorporated into the RNA-induced silencing complex (RISC), where the siRNA gets unwound. The siRNA can now bind to its target mRNA and induce cleavage. **C** Splice-switching ASOs (ssASOs) bind to pre-mRNA and modulate the splicing process. Here, exon 2 is skipped during splicing. Illustration created with BioRender.com.

tolerance to mismatches and thus bear a higher risk of having off-target effects. Additionally, delivery of siRNAs to the target tissue is more challenging due to their double-stranded nature[6].

Knockdown approaches are mainly used to reduce RNA transcript levels and result in reduced protein levels, as needed in the case of toxic gain-of-function (GoF) variants[14,15]. However, other applications, like the reduction of antisense transcripts to enhance the expression of wild-type (WT) alleles, are also possible (see "Haploinsufficiency").

**Splice modulation.** ssASOs target the pre-mRNA and modulate splicing by interfering with the recognition of regulatory splice elements. That means the single-stranded ssASO enters the cell nucleus and binds to its pre-mRNA target in a sequence-specific manner. There it proceeds to block a splice-regulatory element so that it is masked from the splice apparatus. These regulatory elements can be canonical/cryptic splice sites, branchpoints (a *cis*-acting intronic motif localized approximately 18–40 bases upstream of the 3′ ends of the intron), and splicing enhancers and silencers (exonic and intronic)[16]. By hiding splice-regulatory elements from the spliceosome, ssASOs can induce skipping or inclusion of target exons (Fig. 1C). ssASOs are commonly used to restore or disrupt reading frames, leading to increased protein levels for loss-of-function (LoF) variants or decreased protein levels for toxic GoF variants, respectively. Further, ssASOs can be used to increase or decrease transcript levels through different mechanisms. By binding to the untranslated regions, these ASOs can stabilize transcripts or prevent the recognition of upstream open reading frames for increased translation[17,18].

**Table 1 Overview of treatment strategies in relation to pathological mechanism (columns) and pattern of inheritance (rows).**

| | Gain-of-function | Loss-of-function | Dominant-negative |
|---|---|---|---|
| Autosomal dominant | siRNA/gapmer ASO for knockdown ssASO for knockdown or removal of a specific domain | ssASO for reading frame restoration or TANGO to upregulate WT allele | siRNA/gapmer ASO for knockdown ssASO for knockdown or removal of a specific domain |
| Autosomal recessive | *siRNA/gapmer ASO for knockdown ssASO for knockdown or removal of a specific domain** | ssASO for reading frame restoration | *siRNA/gapmer ASO for knockdown ssASO for knockdown or removal of a specific domain** |
| X-linked dominant | siRNA/gapmer ASO for knockdown ssASO for knockdown or removal of a specific domain | ssASO for reading frame restoration** | siRNA/gapmer ASO for knockdown ssASO for knockdown or removal of a specific domain |
| X-linked recessive | *siRNA/gapmerASO for knockdown ssASO for knockdown or removal of a specific domain** | ssASO for reading frame restoration | *siRNA/gapmer ASO for knockdown ssASO for knockdown or removal of a specific domain** |

*Gain-of-function and dominant-negative variants are uncommon in recessive disorders but are mentioned here for completeness.
**TANGO can also be used in X-linked dominant disorders if the gene is expressed from both alleles.
Abbreviations: *ASO* antisense oligonucleotide, *siRNA* small interfering RNA, *ssASO* splice-switching ASO, *TANGO* targeted augmentation of nuclear gene output, *WT* wild type.

## Pathological mechanisms of genetic diseases and their treatment options

While ASOs can also be used to treat somatic variants or infectious diseases[19,20], we focus solely on Mendelian disorders. We thereby distinguish between two main disease mechanisms, namely, GoF and LoF of a protein. Both can be caused by different pathogenic variants, and the possible treatment approaches need to take the genetic background/inheritance pattern of the disease into account (Table 1). Autosomal recessive disorders are mainly associated with LoF variants, while autosomal dominant disorders are associated with different pathological mechanisms. The special case of dominant-negative effects, another type of pathological mechanism, will be briefly discussed at the end of this section.

**Toxic gain-of-function (GoF).** Different types of pathogenic variants can lead to a GoF of a protein. These can be variants that (i) stabilize the protein, leading to overactivation/constitutive activation of the protein, (ii) add a new function to a protein, or (iii) deactivate/cause a loss of inhibitory domains of a protein. Such changes can be toxic for the cell and disrupt its physiological state, ultimately leading to disease. These disease-causing genetic alterations are referred to as toxic GoF variants and are subject to the following discussion.

Pathogenic variants that lead to such toxic effects include but are not limited to duplications or triplications of a gene, missense variants in functional domains that over-activate the protein or add additional functionality or deletions of inhibitory domains or missense variants that inactive inhibitory domains, and expanded repeats that lead to aggregation and sequestration of the mutated protein and binding partners[21].

Therapeutic ASO-based strategies can thus be used to downregulate/knockdown a transcript through gapmer ASOs and siRNAs. ssASOs can be employed in two ways: (i) exon skipping to disrupt the reading frame or (ii) via the modulation of the transcript through the loss (skipping) of crucial domains responsible for the protein's (over)active state.

Since protein dosage is often tightly regulated within the cell[22], it would be important to evaluate to what extent the gene tolerates a downregulation without causing additional harm for knockdown approaches to be safe. However, this evaluation is not trivial. The range of protein levels needed to maintain the physiological state is different for each protein, yet some general considerations can be made with respect to identifying the most suitable treatment approach.

First, a knockdown approach is safe when it is known that a complete loss of the protein (i.e., biallelic LoF variants in a gene) does not alter the physiological state of the cell. This information can be obtained through functional studies in human disease-modeling systems and population databases like gnomAD (https://gnomad.broadinstitute.org/) when homozygous LoF carriers do not show any distinct phenotype. For some proteins, there is a difference in prenatal vs. postnatal knockdown/loss. That means the effect of an embryonic loss of protein function vs. a loss later in life can lead to different—sometimes even opposite—effects[23]. It can also mean that while a prenatal LoF is not tolerated, a postnatal LoF is. Extensive functional analyses are necessary to identify whether a postnatal LoF is tolerated and whether a therapeutic knockdown approach is safe.

Second, if a complete loss of both alleles is not tolerated but the loss of one allele is indicated by healthy heterozygous LoF carriers, a knockdown approach can also be considered. Here, it is crucial to knockdown the protein no more than 50% of the physiological protein activity (this equals the healthy LoF carrier). Since it is very difficult to infer the protein knockdown abilities of an ASO in vivo from in vitro experimental data, an allele-selective RNA therapy that solely targets the mutant allele can be considered. Such an ASO can be designed to bind to the variant site itself or to a single nucleotide polymorphism (SNP) on the same allele as the disease-causing variant[24]. However, specifically targeting SNPs or a defined variant site will limit the design options of the ASO, and one might have to choose less optimal sequences. The more SNPs are available, the better the possibility of identifying efficacious ASOs. The fact that currently none of the approved oligonucleotide therapies for GoF diseases are allele-specific underlines that achieving efficient and specific allele-selectivity is challenging.

Third, using ssASOs, in certain cases, the toxic domain can be spliced out (skipped) from the transcript to allow the production of a protein lacking the toxic domain that is still partially or largely functional. This approach only applies to variants located within in-frame exons. Extensive functional analysis is necessary in such a case to confirm (partial) functionality. Whether a monoallelic or biallelic approach is applicable again depends on the tolerance of the loss of one or both alleles of the gene in question and whether a variant-/allele-specific ASO is efficient.

More complex considerations come into play when toxic GoF variants are identified in genes associated with haploinsufficiency. The term haploinsufficiency describes the situation in which the protein product from one functional allele alone is insufficient to preserve physiological function[25]. Haploinsufficiency is usually

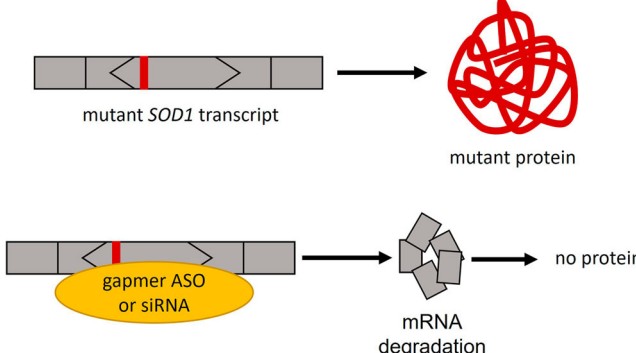

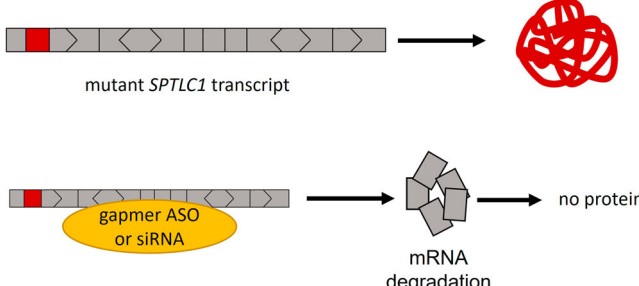

**Fig. 2 ASO-mediated knockdown of mutant *SOD1*.** *SOD1*-associated ALS caused by GoF variants can be therapeutically targeted with gapmer ASOs and siRNAs, eventually leading to mRNA degradation. Less mutant protein will be produced. The frame of the exon is indicated by its shape, exon size is not to scale.

**Fig. 3 ASO-mediated treatment of *SPTLC1* GoF variants.** Toxic GoF variants are located in exon 2 (in-frame exon) of *SPTLC1*. *SPTLC1*-associated ALS can be targeted by knocking down the mutant *SPTLC1* transcript using gapmer ASOs or siRNAs. The frame of the exon is indicated by its shape, exon size is not to scale.

described in connection with LoF variants, but there are reported cases of GoF variants in genes known to be haploinsufficient[26]. This is for example the case for some types of developmental and epileptic encephalopathies. Heterozygous GoF variants in the genes *SCN2A* and *SCN8A* cause disease, as do heterozygous LoF variants associated with haploinsufficiency, albeit with different clinical presentations[27,28]. For the toxic GoF variants in these genes, a knockdown approach using gapmer ASOs, siRNA, or ssASOs should be considered with extreme caution as it bears the risk of pushing an individual from the GoF-associated disease into the LoF-associated condition. Instead, using ssASOs to skip an exon containing the mutated domain while still maintaining part of the protein's function might be a worthy alternative. Again, extensive functional analyses are necessary to show that an ASO therapy can ameliorate the phenotype without causing additional harm.

Notably, knockdown approaches for toxic GoF variants in haploinsufficient genes can be a valid consideration when the LoF phenotype is less severe than the GoF phenotype, and the individual's quality of life can be markedly improved[26].

In summary, GoF variants can be targeted with gapmer ASOs, siRNAs, and ssASOs. Knockdown of target transcript can be achieved in an allele-selective and non-selective manner depending on the loss-of-function tolerance of the gene postnatally.

*Examples of GoF variants for knockdown approaches*
SOD1-associated amyotrophic lateral sclerosis: Heterozygous pathogenic variants in *SOD1* are associated with an adult-onset form of amyotrophic lateral sclerosis (ALS). ALS is a severely debilitating neurodegenerative disease mainly affecting motor neurons, causing early death[29]. *SOD1*-associated ALS cases account for 2–4% of sporadic ALS and up to 20% of familial ALS[30]. ALS caused by *SOD1* variants is highly eligible for an ASO treatment since pathogenic variants in *SOD1* cause a toxic GoF of the protein, and downregulation of the transcript using gapmer ASOs or siRNA are thus useful therapeutic strategies (Fig. 2). First preclinical studies were conducted in 2004, showing that *SOD1* knockdown approaches are possible[31,32]. A first-in-human clinical phase I trial was completed in 2013[29]. The compound, now known as Tofersen (BIIB067), has been studied in a randomized controlled phase III clinical trial[33], and the FDA approved the drug in April 2023[4,34]. The drug was approved based on reducing the plasma biomarker neurofilament light chain, whose levels are strongly correlated with ALS progression. Despite the phase III clinical trial not demonstrating improvement of the clinical endpoints, other studies report stabilization of

patients upon Tofersen treatment[35]. The market approval of Tofersen is an interesting case to study since the drug is associated with a high incidence of treatment-related adverse events[34]. This highlights the need to assess treatment risks, treatment benefits, and disease severity for each individual disease.

SPTLC1-associated amyotrophic lateral sclerosis: Heterozygous pathogenic variants in *SPTLC1* are associated with childhood ALS[36]. These variants are toxic GoF variants resulting in increased enzymatic activity. At the same time, heterozygous LoF of *SPTLC1* is well tolerated; thus, an allele-specific knockdown approach is a possible therapeutic approach for these GoF variants (Fig. 3). Indeed, Mohassel and colleagues were able to show that an allele-specific gapmer ASO targeting the mutant *SPTLC1* transcript is a suitable therapeutic option to reduce the protein function to near-normal levels in vitro.

Notably, exon-skipping therapy would not be an option in this case. The reported GoF variants are all found in exon 2. Since exon 2 is an in-frame exon, one therapeutic consideration could be to skip the exon entirely. To assess whether this leads to a beneficial effect, the function of any domains located within this exon must be considered. The domain that would be lost by the removal of exon 2 from the mRNA is an inhibitory domain, which will result in increased protein activity. Thus, skipping exon 2 is not a viable therapeutic option.

Evidently, one of the patients from the initial publication carries a splice-variant in exon 2, causing an in-frame skipping of exon 2. The individual is fully affected by the disease[36], providing additional evidence that exon-skipping cannot be applied. The toxic GoF variants in *SPTLC1* are an example of why it is important to evaluate the functional consequences of a variant in detail before deciding on the suitability of a therapeutic approach.

ACTL6B-associated neurodevelopmental disorders: Two distinct neurodevelopmental disorders are associated with pathogenic variants in the *ACTL6B* gene[37]. Biallelic LoF variants cause a severe autosomal recessive disorder, whereas presumptive heterozygous GoF variants cause a milder autosomal dominant phenotype. Restoration of the reading frame with the use of ssASOs is the sole therapeutic option available for the autosomal recessive disorder, while multiple approaches can be applied for the treatment of the autosomal dominant disorder.

Generally, for the toxic GoF variants, a knockdown approach can be considered. Since heterozygous LoF carriers are healthy individuals, the gene tolerates downregulating one allele. Using allele-selective gapmers, siRNAs, or ssASOs for a knockdown approach is thus a viable therapeutic option. Additionally, an exon-skipping approach could be considered. The recurring GoF

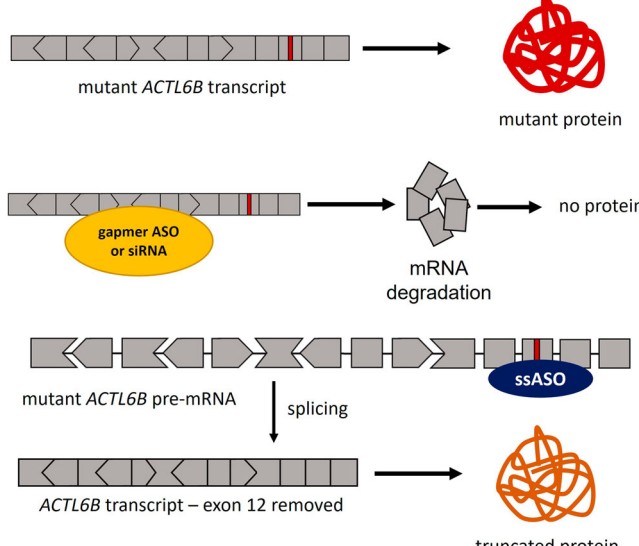

**Fig. 4 ASO-mediated treatment for *ACTL6B* GoF variants.** The mutant *ACTL6B* transcript can be knocked down using gapmer ASOs or siRNAs. A recurring variant in exon 12 (NM_016188.4:c.1027G>A) can be tackled with a ssASO that skips exon 12 to produce a truncated protein. The frame of the exon is indicated by its shape, exon size is not to scale.

variant NM_016188.4:c.1027G>A (p.Gly343Arg)[37] is located within an in-frame exon (exon 12) that could be skipped to possibly produce a partially functional protein (Fig. 4). This approach does not have to be allele-selective and gives therefore more opportunity to design efficacious ASOs. As there is no evidence yet that the truncated protein is functional, confirming functionality in preclinical studies as part of the therapeutic suitability consideration is crucial.

**Loss-of-function (LoF)**. Different genetic alterations can ultimately lead to the LoF of a protein, such as (i) variants disrupting the reading frame (indels or splice variants causing a frameshift), (ii) variants that leave the reading frame intact but destroy regulatory/functional domains of the protein (in-frame indels, missense variants at important amino acids, etc.) or (iii) variants that cause a premature stop of translation (nonsense variants). These genetic changes can result in nonsense-mediated decay of RNA transcripts, the production of unstable proteins, or proteins that are expressed but have no or diminished function[21]. The appropriate ASO approach in such instances will depend on the type of disease-causing variant and the inheritance pattern of the disease, i.e., dominant or recessive disorders.

In general, therapeutic approaches for the LoF of proteins can be restoration of the reading frame to either the canonical transcript or to a modified transcript producing a (partially) functional protein. This is mainly achieved via splice modulation. In the case of autosomal dominant disorders, where an intact WT allele is still present, protein translation from the WT allele can be upregulated. Multiple approaches using splice-modulation or even knockdown of so-called antisense transcripts are possible for the latter situation.

For splice-modulation of LoF variants, determining the disease-causing mechanism is particularly crucial, hence, the possible types of variants and potentially suitable therapeutic strategies for each are covered in more detail in the paragraphs below.

*Cryptic splice site variants*. Cryptic splice sites are splice sites that occur naturally within pre-mRNA transcripts, are infrequently used by the splicing apparatus, and are not normally used for canonical splicing. These cryptic splice sites can be located either within exons or introns. Cryptic splice sites can be introduced or activated through different mechanisms, e.g., the deletion of a silencer element or a variant that increases the strength of the cryptic splice site[38]. The cryptic splice site will then be strengthened up to the point that the site is recognized as a true splice site, and non-coding regions will be included in the mRNA, or parts of the canonical exons excluded. The use of cryptic splice sites often leads to a frameshift and an early stop, abolishing protein production or function.

When cryptic splice site variants are located within an intron, they can result in the inclusion of part of the intron which is then termed a cryptic exon (Fig. 5). These variants are ideal targets for ssASOs since targeting the cryptic splice site variants will restore the normal transcript, and result in the translation of the canonical protein[39]. There are also exonic cryptic splice site variants for which the reading frame can be restored using a ssASO[40]. The design and development of such therapies are more challenging as ASOs targeting exonic cryptic splicing often cause exon skipping of the entire exon rather than including the entire canonical exon.

An example of a cryptic splice site variant: *CEP290-associated Leber congenital amaurosis type 10* Leber congenital amaurosis is an autosomal recessive disorder leading to a progressive loss of vision in childhood. The recurrent, deep-intronic, pathogenic variant NM_025114.4:c.2991+1655A>G in *CEP290* causes a cryptic splice site within intron 26 of the transcript that leads to cryptic exon inclusion with a premature stop codon[41]. This variant is an ideal target for a ssASO (Fig. 5). Indeed, an ASO has been developed that can correct the splicing at the c.2991+1655A>G variant site, and the target drug has been studied in a phase I/II clinical trial[42].

*Canonical splice site variants*. It is not possible to target variants disrupting canonical splicing with ssASOs. These variants include pathogenic variants at the canonical splice sites or in close proximity to the consensus splice sites and variants disrupting the branchpoint. Often, these variants lead to (partial) exon skipping. Once the canonical splice sites are destroyed, the splice apparatus cannot recognize them, so splicing at this site is not possible anymore. As such, a destroyed canonical splice site cannot be repaired using ssASOs.

Of note, if a canonical splice site variant leads to exon-skipping of an entire out-of-frame exon, skipping of an adjacent exon that is also out-of-frame can be considered as a potential therapeutic option. Such an approach has the potential to restore the reading frame of the spliced transcript. These cases can be seen as an out-of-frame exon deletion similar to the prime example of Duchenne muscular dystrophy (DMD). Here, protein function must be assessed and studied as outlined in the sections below.

*Exonic variants*. Exonic variants that lead to LoF of a protein can be targeted with ssASOs, although the considerations are more complex. For more information on what type of exonic variants are amenable to which ssASO approach, we have developed a set of practical guidelines[43]. Splice modulation can be used for skipping in-frame exons that contain pathogenic variants that cause an early stop (nonsense variants and small indels), or it can be used to restore the reading frame for out-of-frame deletions as developed for DMD[44]. In either case, a truncated protein will be produced, which needs to be carefully assessed for maintenance of (partial) functionality. In some diseases, truncated transcripts will cause a milder disease, as is the case for Duchenne and Becker muscular dystrophy. In other conditions, the protein produced after exon skipping is not functional. Evidence for such

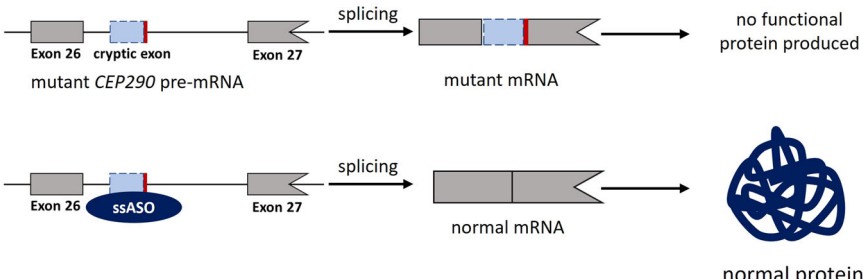

**Fig. 5 ASO-mediated treatment of a deep intronic variant in the *CEP290* gene.** The deep intronic variant NM_025114.4:c.2991+1655A>G leads to the incorporation of a cryptic exon in the *CEP290* mRNA, and no functional protein will be produced. By blocking the variant site with a ssASO, normal splicing will occur, and the physiological protein will be produced. The frame of the exon is indicated by its shape, exon size is not to scale.

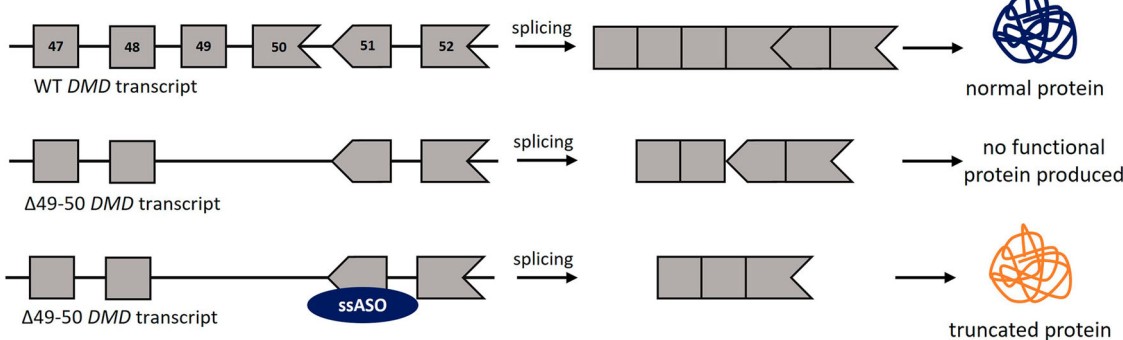

**Fig. 6 ASO-mediated exon-skipping of *DMD* exon 51.** For individuals with exon 49-50 deletions, skipping exon 51 in the *DMD* transcript will restore the reading frame of the mRNA, and an internally truncated but partially functional protein will be produced. The frame of the exon is indicated by its shape, exon size is not to scale.

a case can be a reported deletion of an exon as disease-causing in an affected individual. However, this evidence is often not available and functional studies are needed to confirm protein function after exon skipping strategy has been applied.

Whether ASOs developed for these exonic variants should be allele-selective depends on the variant and inheritance pattern. For recessive diseases, non-allele selective ASOs can be designed. If the variant is homozygous, both alleles will benefit from the treatment. In the case of compound heterozygosity, only one allele will benefit. If it is a dominant disease, the LoF variant is heterozygous and allele-selective approaches are needed where the ssASO can negatively impact the intact WT allele. An example would be a heterozygous disease-causing deletion of an out-of-frame exon that can be corrected by skipping the adjacent upstream or downstream out-of-frame exon to restore the reading frame. Skipping these exons in the non-affected (WT) allele will itself cause a frameshift; here, only an allele-selective approach should be considered.

Examples of exonic LoF variants: *Duchenne muscular dystrophy* DMD is an X-linked disorder caused by LoF variants in the *DMD* gene (coding for the protein dystrophin) and mainly affects males[45]. It is a progressive, muscle-wasting disease leading to early death. *DMD* is the largest known human gene, consisting of 79 exons. Many affected individuals carry deletions that disrupt the *DMD* reading frame and cause an early stop, giving rise to a severe DMD phenotype. Interestingly, deletions that maintain the reading frame lead to a milder form of the disease called Becker muscular dystrophy (BMD).

One potential therapeutic option to treat this condition is the use of ssASOs, with the aim of allowing DMD patients to make BMD-like dystrophins, by restoring the reading frame of the

*DMD* transcript through skipping of additional exons (Fig. 6). Although ASO delivery to muscle is currently hampered by limited uptake, the genetics underlying the disease and possible treatment option with ssASOs make the disease an ideal example to discuss here. Exon-skipping therapies for exons 44, 45, 51, and 53 have been developed and approved by the FDA[45]. This was based on the restoration of very low amounts of dystrophin, and it is still yet to be established whether this strategy is sufficient to slow down disease progression. For more details, we refer the interested reader to previous work focusing on this topic[45–48].

*Tay–Sachs disease.* Tay–Sachs disease is one of the more common autosomal recessive disorders. It manifests in early childhood with the onset of seizures, loss of developmental milestones, hearing loss, and early death[49]. Different genetic variants in *HEXA* are known to cause the disease, and some recurrent (founder) variants have been described. One is a 4-bp insertion (NM_000520.6:c.1274_1277dup, p.Tyr427fs) previously described in the Ashkenazi Jews and Cajuns (Lousiana French ethnicity)[50,51]. Thus, skipping the exon containing this recurrent variant should be considered a therapeutic approach. Unfortunately, the variant is located in an out-of-frame exon, and skipping the exon will lead to the generation of an early stop codon (Fig. 7). There is currently no possibility of treating individuals with this founder variant via a ssASO.

*Haploinsufficiency.* In the case of LoF variants in haploinsufficient genes, there are additional possibilities to employ ASOs. Here, the LoF of one allele means that the remaining WT allele does not produce enough functional protein to maintain a healthy state of the cell/body. While pathogenic variants in the mutant allele can be targeted with the aforementioned approaches, there is also the option to upregulate the expression of the WT protein to

ameliorate the phenotype. These approaches can be summarized as targeted augmentation of nuclear gene output (TANGO), which have successfully been applied in preclinical studies, for example, reducing the seizure frequency in a Dravet syndrome mouse model (Fig. 8)[52–54]. TANGO uses naturally occurring alternative splicing events that result in the generation of non-coding transcripts. By modifying the splicing of these alternative transcripts to increase levels of coding transcript, the protein levels can be raised. Using ssASOs to skip so-called poison exons, whose inclusion will lead to an early stop, or enhance the use of alternative 5′ and 3′ splice sites, are two approaches to increase protein levels for the treatment of genetic diseases associated with haploinsufficiency.

As mentioned earlier, the dosage of a protein needed to sustain physiological function differs for individual cells, individual target tissue, and even throughout different stages of human development. This needs to be considered for assessing if a splice correction can rescue the disease phenotype. Identifying the expression levels of the target transcript in the target tissue via publicly available datasets can give an idea of the feasibility of this approach. Different groups have published lists of genes for which a TANGO approach would be applicable[53,55]. With advanced sequencing techniques, more of these alternative, non-productive transcripts can be discovered for specific tissues and employed as therapeutic strategies.

Other options to increase expression of the WT allele, which are being tested (pre-)clinically, are stabilization of the transcripts through modification of the untranslated regions using ssASOs[17,18] or knockdown of antisense transcripts via gapmer ASOs or siRNAs. This latter strategy is currently being tested in different phase I/IIa clinical trials for Angelman syndrome (ClincalTrials.gov, Identifier: NCT05127226, NCT04259281). This approach makes use of the existence of a natural antisense transcript (NAT) transcribed from the opposite strand of the gene of interest[56]. Transcription of the NAT can, for example, reduce/block transcription of the sense transcript. By degrading the antisense transcript through the use of siRNAs or gapmer ASOs, more of the sense transcript will be produced, increasing the protein levels[57].

*An example of a haploinsufficiency disorder: TANGO approach for SCN1A-associated developmental and epileptic encephalopathy (Dravet syndrome)* Dravet syndrome is caused by heterozygous,

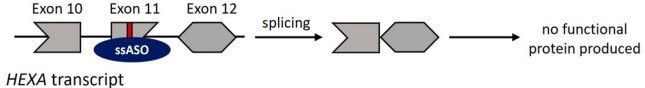

**Fig. 7 ASO-mediated exon-skipping of *HEXA* exon 11.** Skipping exon 11 in the *HEXA* transcript containing the recurrentcNM_000520.6:c.1274_1277dup variant will produce an out-of-frame mRNA transcript and no functional protein. The frame of the exon is indicated by its shape, exon size is not to scale.

pathogenic LoF variants in the sodium channel *SCN1A* associated with haploinsufficiency. Dravet patients present with seizures often refractory to antiepileptic drugs, cognitive decline, and early mortality[58]. As there is an abundance of non-productive *SCN1A* transcripts containing a poison exon between exons 20 and 21, Dravet syndrome is well suited for a TANGO approach (Fig. 8). Skipping of this poison exon was recently shown to increase the canonical (WT) transcript and rescue the phenotype of a Dravet mouse model[46,47]. This approach is currently being evaluated in a clinical phase II trial (ClinicalTrials.gov, Identifier: NCT04740476).

**Dominant-negative effect.** Other than LoF and GoF variants, a third type of pathological mechanism of disease is caused by variants that lead to a dominant-negative effect. Dominant-negative effects occur when the mutant gene product negatively impacts the WT gene product. Although only one allele is mutated, the result leads to a functional loss of over 50% of the protein[21]. The ideal approach in such an instance would be the complete knockdown/degradation of the mutant protein via gapmer ASOs, siRNA or ssASOs. This strategy would avoid any further negative interactions between the mutant and WT alleles. It is important to remember that knockdown approaches are not 100% successful, and leftover mutant protein might still exhibit a dominant-negative effect causing the disease phenotype. Furthermore, this strategy only applies to a gene tolerant to heterozygous LoF or where haploinsufficiency leads to a milder phenotype. Increasing the relative amount of WT protein may have therapeutic effects for some gene products, in which a partial knockdown may be considered clinically beneficial. Similarly, increasing the expression of WT protein, e.g., with TANGO, might be considered therapeutic. However, in both cases, this relies on achieving the desired effect in an allele-specific manner, which is currently technically challenging. As for cases of known dominant-negative effects, extensive functional analyses are necessary to establish how much of the mutant transcript needs to be degraded to have a positive effect on protein function overall.

*An example of a dominant-negative effect. Osteogenesis imperfecta* Osteogenesis imperfecta (OI) is a group of genetic skeletal dysplasias characterized by a wide range of phenotypes, including repeated low-trauma fractures, low bone mass, scoliosis, and short stature. OI is mainly associated with pathogenic variants in genes encoding for collagen type I, such as *COL1A1* and *COL1A2*[59]. Pathogenic variants in these genes can lead to a LoF or a dominant-negative effect, whereby the latter causes a more severe phenotype. The abnormal protein gets incorporated into the collagen triple helices, negatively influencing the structural integrity of the bone[60]. Different ASOs, mainly siRNAs, have been studied to selectively knockdown the mutant alleles in preclinical studies[61–63]. ssASOs to bypass the variants via exon

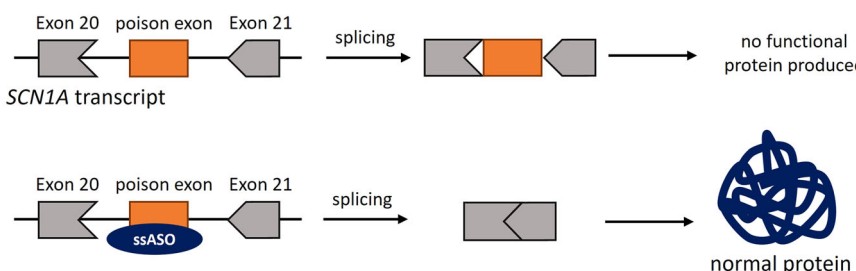

**Fig. 8 TANGO approach in *SCN1A*-associated Dravet syndrome.** A poison exon located in intron 20 of the *SCN1A* transcript when incorporated into the mRNA transcript leads to degradation. Blocking the poison exon with a ssASO prevents its incorporation into the mRNA, and the canonical protein can be produced. The frame of the exon is indicated by its shape, exon size is not to scale.

skipping are not a viable approach for *COL1A1* and *COL1A2*, as the deletion of in-frame exons often leads to severe phenotypes through disruption of trimer assembly[60]. Further clinical investigations into these compounds are currently hampered by the limited delivery of ASO-based drugs to the bone[59].

**Other approaches**. It is also possible to use ASOs that do not target the disease-causing genetic defects but target copy genes or other genes within the disease-causing pathways.

*An example of a copy gene*. A prime example of targeting a copy gene is Nusinersen (Spinraza), which is used for treating 5q SMA. It is one of the best-known and widely used ssASOs to date. SMA is the second most common recessive disorder and is a severely progressive, neurodegenerative disease. It is still described as the leading cause of death in early childhood by a monogenic disease, with most cases not surviving beyond 2 years of age[64]. However, since the discovery of the causative gene *SMN1* in 1995, three disease-modifying therapies — one of which is the ASO Nusinersen - have received marketing authorization from competent authorities worldwide, slowly overthrowing the conception that SMA is a lethal childhood disorder.

SMA is caused by biallelic pathogenic LoF variants in the *SMN1* gene. Yet, the ssASO Nusinersen acts on the *SMN2* transcript. *SMN2* is a copy gene of *SMN1*, and differs from *SMN1* by only a few nucleotides. Both genes encode for the SMN protein. These genetic differences lead to the skipping of exon 7 in the majority of the *SMN2* transcripts, only producing a small percentage of functional SMN protein. While healthy individuals produce sufficient amounts of SMN from *SMN1*, affected individuals cannot produce enough SMN as they rely solely on the pseudogene *SMN2*. Nusinersen was designed to target a splice regulatory element in intron 7 of the *SMN2* transcripts, increasing the incorporation of exon 7 to produce higher protein levels of SMN[65]. The drug has proven to be so efficacious that with early treatment initiation, it is not only able to halt disease progression, but treated individuals are able to develop further and gain developmental milestones not seen before[66,67].

*An example of pathway interaction*. One drug interacting within a disease-causing pathway is Inclisiran (Leqvio) for the treatment of heterozygous familial hypercholesterolemia and atherosclerotic cardiovascular disease. Both disorders can be caused by pathogenic variants in different genes, yet for their treatment, one single drug is available. Inclisiran is a siRNA that targets mRNA of the convertase subtilisin/kexin type 9 (PCSK9) to decrease cholesterol levels[68]. PCKS9 is a liver-secreted enzyme that causes the degradation of LDL receptors, consequently leading to increased LDL levels in the blood. Through downregulation of PCSK9 with a siRNA, circulating LDL levels can be decreased[68]. Other than the previously discussed examples of rare diseases, Inclisiran can be used to target more common disorders.

**Further considerations**
Despite the genetics of each disease, further considerations apply for a successful therapeutic intervention using ASOs.

**Target tissue and disease phenotype**. Currently, the tissues to which ASOs can be delivered in sufficient concentrations are limited. Safe, local delivery can be achieved to the brain and spinal cord via an intrathecal injection or the retina through an intravitreal administration[2]. The liver hepatocytes can be targeted through systemic administration of trimeric N-acetyl galactosamine (GalNac) conjugates. GalNac is a receptor ligand that can bind to the asialoglycoprotein receptor and can be conjugated to different types of ASOs[69]. Efficient delivery to most other tissues is currently not possible, e.g., muscle[70]. While recent years have already brought advances in the stability and binding affinity of ASOs[3], for the treatment of other tissues/organs and to increase the range of diseases eligible for RNA therapies, new delivery methods must be developed[2]. These new methods will have to address the stability and pharmacogenetic profile of the compounds as well as the target affinity and improve cell-penetrating capabilities.

**Time of intervention and expected clinical benefit**. The time point to initiate treatment for a given disease is essential to consider. For neurological disorders, the often-used phrase "time is neurons" is especially meaningful as destroyed tissue/cells cannot be recovered. As such, it is not likely that lost function will be regained. This means that developers of ASO treatments must openly communicate to prospective recipients that these treatments are not a cure. It further implies that diseases where damage is present at birth, i.e., congenital disorders, are generally not good candidates for ASO treatments. Exceptions can be made for conditions that progress postnatally or where a debilitating aspect of the disease (e.g., epilepsy) can be treated and where benefit may be expected upon therapeutic intervention. Further, treatment *in utero* is not yet possible and would require knowledge of the disorder and availability of drugs prenatally, which is currently an unlikely scenario for most rare diseases.

Assuming that for a given disease, a functioning ASO is available, and the delivery to the target tissue is possible, early intervention is essential. Treatment even before symptom onset should be considered. For SMA, pre-symptomatically treated individuals have great clinical benefits and remain stable for longer periods[67]. However, it is important to communicate to the families that the individuals are not cured and that the pathogenic variant is still present. That means the risk of heritability of the disease to the next generation still persists.

Identifying a clinical benefit for progressive disorders, namely to halt or slow down disease progression, is relatively straightforward. It is more complicated to define outcome measures for non-progressive or slowly progressive diseases. For the large group of neurodevelopmental disorders, the question of whether individuals can benefit from postnatal treatment has been an ongoing discussion[71,72]. Therapy can be considered if the genes associated with neurodevelopmental disorders are also needed for neuromaintenance. Moreover, due to the neural plasticity of the brain, neurodevelopmental disorders could potentially benefit from the right therapeutic approach[73]. These considerations need to be made for each disorder individually, and a window of opportunity for therapeutic intervention will have to be established.

To define clinical benefits and establish measures to assess clinical outcomes, the affected individual and their families should be consulted. Measuring biomarkers and testing enzymatic activity alone is not sufficient if the overall benefit of the therapy does not positively impact the recipient's quality of life. There should always be a balance between the disease in question, the treatment burden, and the expected clinical outcome following treatment.

Finally, it should also be evaluated which other therapeutic strategies for a specific disease are available or currently under investigation and what will eventually be the best strategy or strategies for the patients. Some suggestions include gene replacement therapies and gene editing efforts (i.e., CRISPR/Cas9)[74,75], as well as mRNA therapies or enzyme replacement therapies[76,77]. When deciding on the suitability of the genetic approach, the benefits and risks of each approach should be

carefully evaluated. Such discussions should take into account, for example, the frequency of administration, invasiveness of administration procedures, risk of adverse events, efficacy of therapeutic intervention, and clinical benefit.

## Individualized genetic treatments

ASO-based therapeutics can also be used for individualized treatments, as was first shown by the development of Milasen, a ssASO for the treatment of Batten's disease in a single patient[11]. This case opened new possibilities to employ ASOs to target private pathogenic variants. While the general considerations for developing ssASOs also apply to single cases, individualized treatments impose additional considerations and challenges that we would like to touch upon here.

For n-of-1 cases, it is crucial to assess if the patient is suffering from an amenable disease (i.e., matching phenotype and target tissue) while at the same time, the specific variant needs to be eligible for an ASO approach. Patient and variant selection should be evaluated in a structured, objective, and standardized manner. Current international efforts are establishing frameworks to standardize these selection procedures (i.e., the European collaboration 1M1M and the global N=1 Collaborative: https://www.n1collaborative.org/)[78]. Since extensive preclinical and especially clinical testing is not possible for these personalized cases, developing individualized drugs is a race against time. The chemistries and target tissues should be carefully considered. To ensure patient safety and limit adverse events, approved and well-studied chemistries should be used for n-of-1 cases[78]. Global networks and collaboratives have been working to develop guidelines for individualized treatments and recommendations for the preclinical development of ssASOs have been published[79].

For these individualized genetic treatments, clinical outcome measures and potential clinical benefits should be defined in advance and tailored for each case. As for many nano-rare diseases, where no natural history or limited case reports are available, considerations of ASO therapy suitability will be complex and will need to involve an interdisciplinary team of patients and their families, clinicians, ethicists, and researchers.

Clinical and research teams developing genetic treatments should carefully consider their responsibilities towards the affected individual for n-of-1 cases. The possibility of receiving an experimental drug may raise unrealistic hopes in patients and their families. Hence, assessing whether an individual is eligible for an n-of-1 or n-of-few approach should be done cautiously, taking into account the technical possibilities and limitations, the timing of development, the patient's disease status, and potential disease progression. Only if, at the time of assessment, there is sufficient knowledge and evidence that an individual can benefit from n-of-1 treatment, patients and their families should be given hope. However, if treatment would require the development of new chemistries, delivery to hard-to-target tissues, the use of multiple compounds, or the potential benefit does not outweigh the treatment risks, then all relevant information regarding setbacks should be communicated to patients and their families transparently.

Since the publication of Milasen, additional cases have been treated with individualized ASOs, albeit the number being around 15. Every case provides invaluable data and information to advance these n-of-1 treatments and allows more patients to be treated over time.

## Conclusion

There are ample opportunities for ASOs to be used as disease-modifying treatments for monogenic disorders. Limitations for the broad application of ASOs include restricted delivery options to the affected tissues and the current lack of understanding of the required level of upregulation or downregulation of a specific protein to rescue a disease phenotype. Advances in technology will broaden the types of disorders that will be treatable with ASOs in the future. ASO strategies should be specifically tailored to the disease mechanism, disease type, and suitability of the prospective individual receiving the intervention. While there should be a balance between disease and treatment burden, in the end, the expected clinical benefit should always outweigh the therapeutic risks.

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

## Acknowledgements

The authors would like to thank their collaborators, especially from the Dutch Center for RNA Therapeutics, the TAILORED consortium, 1M1M, the N=1 Collaborative, and the many patients and patient organizations with whom we are in continuous discourse for helpful discussions that led to the idea of this paper. The DCRT-LUMC is funded by the Department of Human Genetics at the Leiden University Medical Center. M.C.L. is a

Walter Benjamin Fellow funded by the German Research Foundation project 521414448. A.A.R. and Wv.R.M. are funded by ZonMW PSIDER project 10250022110002.

## Author contributions

M.C.L., Wv.R.M., and A.A.R. conceptualized the work. M.C.L. and A.A.R. wrote the paper. M.C.L. prepared the figures. W.v.R.M. edited the paper.

## Competing interests

A.A.R. is a steering committee member of the N = 1 collaborative. A.A.R., Wv.R.M., and M.C.L. are members of the N=1 Collaborative working groups on preclinical development and patient identification. A.A.R. discloses being employed by LUMC, which has patents on exon skipping technology, some of which has been licensed to BioMarin and subsequently sublicensed to Sarepta. As co-inventor of some of these patents, A.A.R. is entitled to a share of royalties. A.A.R. further discloses being an ad hoc consultant for PTC Therapeutics, Sarepta Therapeutics, Regenxbio, Alpha Anomeric, Lilly BioMarin Pharmaceuticals Inc., Eisai, Entrada, Takeda, Splicesense, Galapagos and Astra Zeneca. Past ad hoc consulting has occurred for CRISPR Therapeutics, Summit PLC, Audentes Santhera, Bridge Bio, Global Guidepoint, and GLG consultancy, Grunenthal, Wave, and BioClinica. A.A.R. also reports having been a member of the Duchenne Network Steering Committee (BioMarin) and being a member of the scientific advisory boards of Eisai, hybridize therapeutics, silence therapeutics, and Sarepta therapeutics. Past SAB memberships: ProQR, Philae Pharmaceuticals. Remuneration for these activities is paid to LUMC. LUMC also received speaker honoraria from PTC Therapeutics, Alnylam Netherlands, Pfizer, and BioMarin Pharmaceuticals and funding for contract research from Italfarmaco, Sapreme, Eisai, Galapagos, Synnaffix, and Alpha Anomeric. Project funding is received from Sarepta Therapeutics and Entrada. W.v.R.M. discloses being employed by LUMC, which has patents on exon-skipping approaches for neurological disorders. In the past, some of these patents have been licensed to ProQR therapeutics. As co-inventor of these patents, W.v.R.M. is entitled to a share of milestone payments and royalties. W.v.R.M. further discloses being an ad hoc consultant for Accure Therapeutics and Herbert Smith Freehills. Remuneration for these activities is paid to the LUMC. LUMC also received funding for contract research from UniQure and Amylon Therapeutics.

## Additional information

## N = 1 Collaborative

Marlen C. Lauffer [1], Willeke van Roon-Mom[1] & Annemieke Aartsma-Rus[1]✉

A full list of members and their affiliations appears in the Supplementary Information.

