## [Peer Review File · Communications Medicine]

Reviewers' comments:

Reviewer #1 (Remarks to the Author):

COMMSMED-23-0346 – Review

The manuscript by Aartsma-Rus and cols provides a detailed but simple (i.e. suitable for non-expert readers) analysis of ASO therapies for Mendelian diseases. It highlights the main ASO strategies that can be used to rescue for different types (gain-of-function and loss-of-function) mutations.

Organization of the content is logical and factually sound. The proposed figures are very good and self-explicative. However, I consider that some attention (even if brief) should be given to some technical aspects of translating ASO strategies to the clinic (see below comments nr. 10 and 11). The language and grammar are of good quality, however there are some typos or missing words along the text that need to be fixed (some examples are given below, but the list not exhaustive, the whole text should be carefully revised).

1. P.4, last parag – the authors make a clear mechanistic separation between variants leading to 'gain-of-function' (GoF) of a protein or the 'loss-of-function' (LoF) of a protein which is perfectly fine. However, because the manuscript deals with approaches aimed at treating Mendelian disorders, they should add that these situations are usually related to recessive (GoF) and dominant (LoF) diseases/variants.

2. p.5, parag. 4 – the latter comment also applies here, besides this paragraph should be under LoF and not GoF.

3. p.5, 1st parag – under the description of 'gain-of-function' variants, although it is mentioned that one possibility is that the variant adds a function to the protein, it is insufficiently clear that among the most frequent situations are those in which the new 'acquired' function is that it results in a protein that is toxic, thus harmful, to the cell/organism. This should be further elaborated as it is the most frequent situation for GoF variants.

4. p.5, 3rd/4th parag – the description on genes carrying GoF variants being associated with haploinsufficiency is scarce and quite counterintuitive, it should be further elaborated and at least one example provided or removed from the GoF section and just mentioned in the LoF section.

5. pp.6/8 – in the examples provided for ASO to treat several Mendelian diseases, mention should be given on the mendelian type for each of them (autosomal, sex-linked, recessive, mixed types, etc.)

6. p.9, 1st parag – the first sentence "LoF of a protein can be caused by different ..." is inaccurate as it is the presence of a premature termination codon that leads to NMD and not the other way round, it should be rephrased.

7. p.9, 2nd parag – the term "truncated transcript" doesn't reflect the various possibilities of alternatively spliced transcripts, should thus be rephrased.

8. p.10, 1st parag; pp.11/14 – comment 5 also applies to the examples provide here.

9. Some technical aspects of translating ASO therapies to the clinic are missing. Points to be added include:

a. aspects regarding the enhancement of stability, increase of target affinity, and improvement of the ASO pharmacokinetic profile including mention to novel '3rd generation ASOs', with some of the backbone modifications such as PMO (as in DMD).

b. Novel strategies dealing with enhancement of transfection efficiency, should be also mentioned conjugating ASOs to cell-penetrating peptides (CPPs).

10. Comparison of advantages/disadvantages of ASO strategies to other ongoing strategies, namely gene editing (CRISPR) or gene/mRNA therapies should be provided, even if just as a table.

Minor

- insertions/deletions is usually abbreviated to indels
- It is now recommended to always use the term 'variants' instead of 'mutations'
- p.1 (and subsequent) – pathomechanism pathophysiological mechanism
- >7,000 rare diseases ~8,000 (see: "Medicines in Development: Rare Diseases. A Report on Orphan Medicines in the Pipeline", 2021)
- p.2 – patients recent recommendations endorse replace by 'individuals'
- p.2 – monogenetic monogenic
- p. 5 (and subsequent) – overactivate over activate
- p.11 – ou-of-frame out-of-frame

Reviewer #2 (Remarks to the Author):

This manuscript has the potential to be an outstanding addition to the literature. There have been many reviews on ASOs/duplex RNAs recently, so editors/reviewers should be wary of simply adding more of the same to a cluttered field. This manuscript, however, fills a unique niche. It begins with a standard, concise background lesson. It then moves onto a description of some of the best work on use of antisense technologies to treat monogenic disorders. I am not aware of any other recent review for this fast moving and impactful area. The authors are intellectual leaders of the field, and this leadership is evident in the breadth of coverage.

I have two primary concerns. The first is that the text describing ASOs and mechanisms is often too brief. I have no expectation that the average reader will understand the nuances, while the expert will not benefit from superficial analysis. Added text at key points would be helpful.

My second concern is the section on personalized genetic treatments. The authors are leaders of this field and I found this text well written and powerful. However, what does it add to this paper? Perhaps the editor might consider asking the authors to write a short Perspective that could include this information, a Perspective could be published alongside the paper that I have been asked to review? (Hopefully, without a second open access fee). That would be a better service to the readers of Nature Communications.

Specific points.

Line 144 I know what the authors are referring to when they describe a "design window" for allele-selective ASOs. I doubt a typical reader will, even one within the oligo therapeutics field. This is a good example of the type of passage where the authors would be well-advised to slow down and make sure that they are writing for a broad audience.

Line 180. The approval of Tofersen is an interesting story. When not engage the reader by spending a paragraph describing it. The Tofersen story directly impacts the "Possibilities and Limitations" of the strategy.

Line 193. Text describing SPTLC1 ASOs. The text is not inaccurate, but I needed to read it several times to understand it. There are several relatively complex concepts that are being fired at the reader one after another. This is another example where a modest re-writing, perhaps using 3-4 short paragraphs, might be helpful.

Line 209. ACTL6B text. The reader is being left to guess about why this section is included because the text is cryptic.

Line 281. I must confess, the text about canonical splice variants was too terse and lost me.

Line 313. The Duchenne MD field is another example of a dramatic human and scientific story, that

reveals much about the nuances of drug development. Nature Communications is a high visibility journal with a broad audience, including readers who may be new to the field. A couple paragraphs telling this story would be welcome.

Line 419. Nusinersen. Same reasoning as with DMD. A great story that should be summarized. Also, how do the authors feel that ASOs will compete, long term, with gene therapy or small molecule drugs? That consideration would also seem to impact "Possibilities and Limitations".

Line 449. I'd be interested in the authors thoughts on lessons we should take from the Ionis/Roche HTT lowering clinical trial.

In summary, I hope the authors view these section as I do - a chance to make straightforward changes that will help tell a more compelling story. Nature Communications would be fortunate to provide a home for an accompanying Perspective on N+1 therapy from this author team.

We would like to thank both reviewers for carefully reading our manuscript. The feedback and recommendations were very useful for improving our work.

The main aim of our work is to educate primarily clinicians working with rare disease patients who often do not have enough information regarding which therapeutic approaches are most useful for which disorder. We would like to close this knowledge gap. Therefore, we have expanded on our text and added more detail for a better understanding of the matter, as suggested by both reviewers.

To keep our focus, we did not elaborate on the technical limitations that still have to be overcome, as well as other therapeutic strategies, as this is outside of the scope of this work.

A point by point rebuttal is found below

Reviewer #1 (Remarks to the Author):

COMMSMED-23-0346 – Review

The language and grammar are of good quality, however there are some typos or missing words along the text that need to be fixed (some examples are given below, but the list not exhaustive, the whole text should be carefully revised).

Author response: Thank you very much, we have carefully checked our text again.

1. P.4, last parag – the authors make a clear mechanistic separation between variants leading to ‘gain-of-function’ (GoF) of a protein or the ‘loss-of-function’ (LoF) of a protein which is perfectly fine. However, because the manuscript deals with approaches aimed at treating Mendelian disorders, they should add that these situations are usually related to recessive (GoF) and dominant (LoF) diseases/variants.

Author response: Thank you for pointing this out; we fully agree. We have added it to our text and also added a table showing the relation between pathophysiological mechanism and mode of inheritance with the respective treatment approach(es). Please see Table 1, and the text now reads as follows:

“Both can be caused by different pathogenic variants, and the possible treatment approaches need to take the genetic background/inheritance pattern of the disease into account (Tab. 1). Autosomal recessive disorders are most commonly associated with LoF variants, while autosomal dominant disorders are associated with different pathomechanisms.”

2. p.5, parag. 4 – the latter comment also applies here, besides this paragraph should be under LoF and not GoF.

Author response: Thank you for your comment. We understand that this paragraph might have been misleading as it talks about LoF within the GoF section. What we tried to explain are the conditions in which a knockdown approach for a GoF variant is safe. This is why we have to discuss loss of function in this context. To clarify, we have changed the sentence structure and re-phrased the paragraph

“First, a knockdown approach is safe when it is known that a full loss of the protein (i.e. biallelic LoF variants in a gene), does not alter the physiological state of the cell, knockdown approaches can be applied.”

3. p.5, 1st parag – under the description of ‘gain-of-function’ variants, although it is mentioned that one possibility is that the variant adds a function to the protein, it is insufficiently clear that among the most frequent situations are those in which the new ‘acquired’ function is that it results in a protein that is toxic, thus harmful, to the cell/organism. This should be further elaborated as it is the most frequent situation for GoF variants.

Author response: We agree that it was not fully clear that the gain of function is toxic to the cell in these cases and have rephrased the paragraph.

“3.1 Toxic Gain-of-function (GoF)

Different types of pathogenic variants can lead to a GoF of a protein. These can be mutations variants that stabilize the protein, lead to overactivation/constitutive activation of the protein, add a function to the protein, or deactivate/cause the loss of inhibitory domains of a protein. Such changes can be toxic for the cell and disrupt its physiological state, ultimately leading to disease. These disease-causing genetic alterations are referred to as toxic GoF variants and subject to the following discussion.”

4. p.5, 3rd/4th parag – the description on genes carrying GoF variants being associated with haploinsufficiency is scarce and quite counterintuitive, it should be further elaborated and at least one example provided or removed from the GoF section and just mentioned in the LoF section.

Author response: Thank you for addressing this point. We fully agree and have now provided examples.

“More complex considerations come into play when toxic GoF variants are identified in genes associated with haploinsufficiency. The term haploinsufficiency describes in which the protein product from one functional allele alone is insufficient to preserve physiological function²¹. Haploinsufficiency is usually described in connection with LoF variants, but there are reported cases of GoF variants in genes known to be haploinsufficient²². This is for example the case for some types of Developmental and Epileptic Encephalopathies. Heterozygous GoF variants in the genes SCN2A and SCN8A cause disease, as do heterozygous LoF variants associated with haploinsufficiency, albeit with different clinical presentations^{23,24}. For the toxic GoF variants in these genes, a knockdown approach using gapmer ASOs, siRNA, or ssASOs should be considered with extreme caution as it bears the risk of pushing an individual from the GoF-associated disease into the LoF-associated condition.”

5. pp.6/8 – in the examples provided for ASO to treat several Mendelian diseases, mention should be given on the mendelian type for each of them (autosomal, sex-linked, recessive, mixed types, etc.)

Author response: Thank you for this suggestion. Our examples were chosen to show how the different ASO approaches can be used in different situations, when they work and when they do not work rather than providing examples for the different types of Mendelian disorders. Going by the different disease mechanisms, we also cover the different inheritance patterns.

To clarify, we have now also put that into our table as discussed for one of your previous comments.

6. p.9, 1st parag – the first sentence "LoF of a protein can be caused by different ..." is inaccurate as it is the presence of a premature termination codon that leads to NMD and not the other way round, it should be rephrased.

Author response: We have restructured the paragraph.

"Different genetic alterations can ultimately lead to the LoF of a protein like variants disrupting the reading frame (indels or splice variants causing a frameshift), variants that leave the reading frame intact but destroy regulatory/functional domains of the protein (in-frame deletions/insertions, missense variants at important amino acids, etc.), or variants that cause a premature stop of translation (nonsense variants). These genetic changes can result in nonsense-mediated decay of RNA transcripts, the production of unstable proteins, or proteins that are expressed but have no or diminished function."

7. p.9, 2nd parag – the term "truncated transcript" doesn't reflect the various possibilities of alternatively spliced transcripts, should thus be rephrased.

Author response: Thank you for this comment, we have now rephrased this sentence.

"In general, therapeutic approaches for these LoF of proteins can be restoration of the reading frame to either the canonical full length transcript or to a truncated modified transcript producing a (partially) functional protein."

8. p.10, 1st parag; pp.11/14 – comment 5 also applies to the examples provide here.

Author response: Please do see our response to comment 5.

Providing examples for all the different types of disorders for all the different ASO approaches is outside of the scope of this manuscript. Instead we make things more concrete by using distinct examples to illustrate our general explanation.

9. Some technical aspects of translating ASO therapies to the clinic are missing. Points to be added include:

- a. aspects regarding the enhancement of stability, increase of target affinity, and improvement of the ASO pharmacokinetic profile including mention to novel '3rd generation ASOs', with some of the backbone modifications such as PMO (as in DMD).
- b. Novel strategies dealing with enhancement of transfection efficiency, should be also mentioned conjugating ASOs to cell-penetrating peptides (CPPs).

We believe these aspects are beyond the scope of this work. Our aim is to inform clinicians and geneticists about the possibilities of ASO therapies and which type of disorders can be treated with which approach. We further wanted to focus on what is possible right now rather than what can become possible in the future, as our discussions with patient representatives and clinicians have highlighted a need for this. Yet, we have added a sentence in the section 4.1 *Target tissue and disease*

phenotype and in our conclusion to address that more opportunities might become available in the future.

10. Comparison of advantages/disadvantages of ASO strategies to other ongoing strategies, namely gene editing (CRISPR) or gene/mRNA therapies should be provided, even if just as a table.

Author response: Thank you very much for this suggestion. While we agree that these are important other strategies, discussing the advantages and disadvantages is outside of the scope of this manuscript. We have added a section on “4.3 Other therapeutic strategies” to raise awareness of other existing therapeutic options and the need to evaluate which one is the best option for each disease.

Minor

- insertions/deletions is usually abbreviated to indels

Author response: Thank you, we have changed it to indels.

- It is now recommended to always use the term ‘variants’ instead of ‘mutations’

Author response: We fully agree and have replaced the word “mutation” with “(pathogenic) variant” throughout.

- p.1 (and subsequent) – pathomechanism pathophysiological mechanism

- >7,000 rare diseases ~8,000 (see: "Medicines in Development: Rare Diseases. A Report on Orphan Medicines in the Pipeline", 2021)

Author response: Thank you for pointing this out, we have changed the number. Depending on the source, different numbers are circulated, but we now went with approx. 8,000.

- p.2 – patients recent recommendations endorse replace by ‘individuals’

Author response: Thank you for making us aware of these new recommendations. We have changed the word throughout the manuscript where applicable.

- p.2 – monogenetic monogenic

Author response: Thank you, this was changed throughout the manuscript now.

- p. 5 (and subsequent) – overactivate over activate

Author response: Thank you, has been changed.

- p.11 – ou-of-frame out-of-frame

Author response: Corrected.

Reviewer #2 (Remarks to the Author):

I have two primary concerns. The first is that the text describing ASOs and mechanisms is often too brief. I have no expectation that the average reader will understand the nuances, while the expert will not benefit from superficial analysis. Added text at key points would be helpful.

Author response: Thank you for your feedback. We have expanded on the explanation of the ASO mechanisms and on other sections as well.

My second concern is the section on personalized genetic treatments. The authors are leaders of this field and I found this text well written and powerful. However, what does it add to this paper? Perhaps the editor might consider asking the authors to write a short Perspective that could include this information, a Perspective could be published alongside the paper that I have been asked to review? (Hopefully, without a second open access fee). That would be a better service to the readers of Nature Communications.

Author response: We were advised not to write a second manuscript on the n-of-1 section. We expanded on the section and stressed why we also include it in this manuscript. We do hope that with the modest changes made, this section now fits better in our story.

Specific points.

Line 144 I know what the authors are referring to when they describe a "design window" for allele-selective ASOs. I doubt a typical reader will, even one within the oligo therapeutics field. This is a good example of the type of passage where the authors would be well-advised to slow down and make sure that they are writing for a broad audience.

Author response: Thank you for your remark. We have reworked this section.

"Such an ASO can be designed to either bind to the variant site itself or to a single nucleotide polymorphism (SNP) on the same allele as the disease-causing variant²⁰. However, specifically targeting SNPs or a variant site will limit the design options of the allele-specific ASO and one might have to choose less optimal sequences. The more SNPs are available, the better the possibilities to identify efficacious ASOs. The fact that currently none of the approved oligonucleotide therapies for GoF diseases are allele-specific underlines that achieving efficient and specific allele-selectivity is challenging."

Line 180. The approval of Tofersen is an interesting story. When not engage the reader by spending a paragraph describing it. The Tofersen story directly impacts the "Possibilities and Limitations" of the strategy.

Author response: Thank you, we do agree that this is an interesting story. We have expanded on the story a bit, yet want to keep the focus of this work on the approaches and considerations without going into detail on market approval and the politics behind it.

Line 193. Text describing SPTLC1 ASOs. The text is not inaccurate, but I needed to read it several times to understand it. There are several relatively complex concepts that are being fired at the reader one

after another. This is another example where a modest re-writing, perhaps using 3-4 short paragraphs, might be helpful.

Author response: We have re-written that section and do hope it adds more clarity.

The authors refrain from posting the modified text here because of its length, and kindly ask the reviewer to check the re-written text directly in the manuscript.

Line 209. ACTL6B text. The reader is being left to guess about why this section is included because the text is cryptic.

Author response: Thank you very much. We included this example since gapmerASO/siRNA, ssASOs for a knockdown as well as ssASOs for exon-skipping of one specific exon containing the GoF variants can be used, which is an unusual case. We have emphasized this more to make this more clear.

The authors refrain from posting the modified text here because of its length, and kindly ask the reviewer to check the re-written text directly in the manuscript.

Line 281. I must confess, the text about canonical splice variants was too terse and lost me.

Author response: Thank you for pointing this out. Being clear and understandable in regard to why canonical splice site variants cannot be corrected with ASOs is one of the most important so we have rewritten this to make it more clear.

"It is not possible to target variants disrupting canonical splicing with ssASOs. These variants include pathogenic variants at the canonical splice sites or in close proximity to the consensus splice sites and variants disrupting the branchpoint. Often these lead to (partial) exon skipping. Once the canonical splice sites are destroyed, the splice apparatus cannot recognize them, so splicing at this site is not possible anymore. As such, a destroyed canonical splice site cannot be repaired using ssASOs.

Off note, if a canonical splice site variant leads to exon-skipping of an entire out-of-frame exon, skipping of an adjacent exon that is also out-of-frame can be considered as a therapeutic option. Such an approach has the potential to restore the reading frame. These cases can be seen as an out-of-frame exon deletion similar to the prime example of Duchenne Muscular Dystrophy (Fig. 6). Here, protein function has to be assessed and studied as outlined in the sections below."

Line 313. The Duchenne MD field is another example of a dramatic human and scientific story, that reveals much about the nuances of drug development. Nature Communications is a high visibility journal with a broad audience, including readers who may be new to the field. A couple paragraphs telling this story would be welcome.

Author response: We agree that this is a very interesting scientific story, however, the authors must admit some bias in stating this. We have expanded on the story, yet referred to previous work which is a better source to have a full picture. We supplemented our text with a figure as well.

The authors refrain from posting the modified text here because of its length, and kindly ask the reviewer to check the re-written text directly in the manuscript.

Line 419. Nusinersen. Same reasoning as with DMD. A great story that should be summarized. Also, how do the authors feel that ASOs will compete, long term, with gene therapy or small molecule drugs? That consideration would also seem to impact "Possibilities and Limitations".

Author response: We have expanded on the nusinersen story. We also added a small section on considering other therapies: *4.3 Other therapeutic strategies*. We have refrained from discussing other options in detail as our focus is on explaining the ASO approaches for rare diseases to the broader audience.

Off note, but to add to this train of thought of the reviewer: Interestingly, despite having the option, parents of SMA affected children in some cases still chose Spinraza over Zolgensma or use a combination. This indicates that the market should provide multiple therapeutics, and it will be interesting to identify the reasons behind such decisions in parents/care givers.

Line 449. I'd be interested in the authors thoughts on lessons we should take from the Ionis/Roche HTT lowering clinical trial.

Author response: This is beyond the scope of this educational piece.

REVIEWERS' COMMENTS:

Reviewer #1 (Remarks to the Author):

The manuscript by Aartsma-Rus and cols has been significantly revised having thus being greatly improved with addition of Table 1 being a major upgrade.

Just a few minor points:

1. Previous comment nr.9a: Although the reviewer understands that future ASO therapeutic options are outside the scope of the current review, it would be useful to the clinical reader to mention some of the '3rd generation ASOs', in CURRENT therapeutic use, i.e., FDA-approved [check PMID: 36881759].
2. Pathomechanism(s): although this was corrected on p.1 the word still appears throughout the text.

Reviewer #2 (Remarks to the Author):

The authors have successfully addressed my concerns.

We would like to thank both reviewers for additional feedback.

Reviewer #1 (Remarks to the Author):

The manuscript by Aartsma-Rus and cols has been significantly revised having thus being greatly improved with addition of Table 1 being a major upgrade.

Just a few minor points:

1. Previous comment nr.9a: Although the reviewer understands that future ASO therapeutic options are outside the scope of the current review, it would be useful to the clinical reader to mention some of the '3rd generation ASOs', in CURRENT therapeutic use, i.e., FDA-approved [check PMID: 36881759].

We added a sentence on the different generations of ASOs in the "Mechanisms of action" section and cited the paper recommended here.

We further added another sentence stating that in the recent years there were already improvements made in the stability and binding affinity of ASOs in the "Further considerations" section.

The PMO for Duchenne was already cited in the DMD subsection.

2. Pathomechanism(s): although this was corrected on p.1 the word still appears throughout the text.

This has now been changed throughout the manuscript.